# An In Vitro Study of Osteoblast Response on Fused-Filament Fabrication 3D Printed PEEK for Dental and Cranio-Maxillofacial Implants

**DOI:** 10.3390/jcm8060771

**Published:** 2019-05-31

**Authors:** Xingting Han, Neha Sharma, Zeqian Xu, Lutz Scheideler, Jürgen Geis-Gerstorfer, Frank Rupp, Florian M. Thieringer, Sebastian Spintzyk

**Affiliations:** 1Section Medical Materials Science and Technology, University Hospital Tübingen, Osianderstr. 2–8, D-72076 Tübingen, Germany; xuzeqian934130@126.com (Z.X.); lutz.scheideler@med.uni-tuebingen.de (L.S.); juergen.geis-gerstorfer@med.uni-tuebingen.de (J.G.-G.); Frank.Rupp@med.uni-tuebingen.de (F.R.); Sebastian.Spintzyk@med.uni-tuebingen.de (S.S.); 2Medical Additive Manufacturing Research Group, Hightech Research Center, Department of Biomedical Engineering, University of Basel, CH-4123 Allschwil, Switzerland; neha.sharma@unibas.ch (N.S.); Florian.Thieringer@swiss-mam.ch (F.M.T.); 3Department of Oral and Cranio-Maxillofacial Surgery, University Hospital Basel, CH-4031 Basel, Switzerland

**Keywords:** polyetheretherketone, dental and cranio-maxillofacial implants, prostheses, fused-filament fabrication, in vitro study, surface modification and characterization, cell adhesion, cell metabolic activity, cell proliferation

## Abstract

Polyetheretherketone (PEEK) is a prime candidate to replace metallic implants and prostheses in orthopedic, spine and cranio-maxillofacial surgeries. Fused-filament fabrication (FFF) is an economical and efficient three-dimensional (3D) printing method to fabricate PEEK implants. However, studies pertaining to the bioactivity of FFF 3D printed PEEK are still lacking. In this study, FFF 3D printed PEEK samples were fabricated and modified with polishing and grit-blasting (three alumina sizes: 50, 120, and 250 µm) to achieve varying levels of surface roughness. In vitro cellular response of a human osteosarcoma cell line (SAOS-2 osteoblasts, cell adhesion, metabolic activity, and proliferation) on different sample surfaces of untreated, polished, and grit-blasted PEEK were evaluated. The results revealed that the initial cell adhesion on different sample surfaces was similar. However, after 5 days the untreated FFF 3D printed PEEK surfaces exhibited a significant increase in cell metabolic activity and proliferation with a higher density of osteoblasts compared with the polished and grit-blasted groups (*p* < 0.05). Therefore, untreated FFF 3D printed PEEK with high surface roughness and optimal printing structures might have great potential as an appropriate alloplastic biomaterial for reconstructive cranio-maxillofacial surgeries.

## 1. Introduction

Additive manufacturing (AM), also known as three-dimensional (3D) printing or rapid prototyping (RP), is a layer-by-layer manufacturing method that fuses or deposits materials—such as metals, ceramics, plastics, liquids, powders, or even living cells—to create specimens [1]. Additive manufacturing is a revolutionary technology characterized by an increased degree of design freedom and the ability to produce complicated geometries that are impossible to achieve with traditional manufacturing processes [2]. In low-volume production cycles, AM results in a less waste generation and enables reductions in both tooling and production costs [3]. Therefore, this technique has the potential to be a pervasive, efficient, and environmentally friendly choice in many sectors (e.g., aerospace, electronics, automotive, and medicine industries) [4].

There are several AM technologies such as fused-filament fabrication (FFF), stereolithography (SLA), selective laser sintering (SLS), multi-jetting, direct metal laser sintering (DMLS), and electron beam melting (EBM) [3,5]. Among these, extrusion-based technologies such as FFF have become one of the most popular and fastest growing manufacturing techniques [6,7]. With less-expensive materials, systems, and relatively lower labor costs, FFF has successfully incorporated a variety of materials such as thermoplastics and metals as raw materials [8]. Additionally, in recent years, this technology has captured the attention of the medical sector, given its ability to process high-temperature thermoplastics such as polyetheretherketone (PEEK) [8].

PEEK, a polyaromatic semi-crystalline thermoplastic polymer (–C_6_H_4_–O–C_6_H_4_–O–C_6_H_4_–CO–)_n_, is an important member of the polyaryletherketone (PAEK) family [6,9]. In recent decades, PEEK has been widely used in several clinical applications: prostheses for fracture fixation, cranio-maxillofacial defects repair, spinal implants, total joint arthroplasty (TJA), and even as suture anchors in soft-tissue repairs [10,11]. Its subsequent applications in dentistry have been substantial and include dental implants, implant-supported bars, dental prostheses, abutments, and clamps [11,12]. PEEK possesses a number of characteristic properties that make it a suitable candidate for replacing metal implant components involved in orthopedics and reconstructive surgeries. PEEK exhibits good biocompatibility, high chemical resistance, low density (1.32 g/cm^3^) and is bioinert [12]. Unlike titanium (Ti) and ceramics implant materials, which have an elastic modulus of 102–110 GPa and 210 GPa respectively, PEEK has a lower elastic modulus (3–4 GPa) closer to the human cortical bone (14 GPa) [10]. This property helps to reduce stress shielding and associated peri-implant bone resorption, which can ultimately result in implant failure [12,13,14,15]. In addition, the modulus of PEEK can be tailored by incorporating other materials such as hydroxyapatite (HA) or carbon fibers to improve its mechanical strength [10,16]. Another significant advantage of PEEK is that it is radiolucent. Unlike metal implants—which are radio-opaque and create artifacts in X-ray radiography, computed tomography (CT), and magnetic resonance imaging (MRI)—PEEK creates no such artifacts [17]. It thereby enables better radiographic evaluation of bone healing [18]. Finally, PEEK has a stable aromatic structure with a melting temperature around 343 °C—it is thermally stable at sterilization temperatures and do not degrade during either gamma radiation or steam sterilization [19]. 

PEEK medical implants have traditionally been manufactured using subtractive manufacturing techniques, such as injection molding [16]. Given the current availability of a wide range of AM technologies, however, efforts to process PEEK are now primarily focused on two primary approaches. Powder bed fusion processes such as SLS and extrusion-based technologies such as FFF have shown great potential [20]. Several successful examples in biomedical applications using SLS-fabricated PEEK appear in the literature [4,6,21]. Although the physical properties of PEEK polymer make it difficult to use in FFF 3D printing, recent technological advancements in extrusion-based systems have made it possible to process PEEK and fabricate patient-specific implants (PSIs) [2,7,22]. The mechanical strength and in vitro cytotoxicity of FFF 3D printed PEEK for medical applications have also been recently evaluated in some studies [4,6,10,21,22]. 

As noted above, PEEK is bioinert. This property impedes its osseointegration potential and hampers its clinical adoption [23]. Currently, two major strategies have been applied to improve the bioactivity of PEEK implants [17]. One approach is to prepare bioactive PEEK composites by incorporating bioactive materials into PEEK substrates, such as titanium dioxide (TiO_2_) and HA. Another strategy involves activating the PEEK surface with coating materials (e.g., bioactive glass, TiO_2_, HA). However, these modifications have some limitations. The bioactive materials in PEEK composites may influence the mechanical strengths of PEEK and lead to stress shielding [24]. On the other hand, PEEK surface coatings may result in insufficient cohesion and delamination, which may lead to local tissue inflammation [24]. Surface topographical modification is another interesting approach to increase the bioactivity of PEEK. Recent studies have indicated that microroughened topographies on PEEK surface can promote ingrowth of both soft and hard tissues [25]. Despite the attractive advantages and progresses made in improving the bioactivity of PEEK implants, the biological performance of FFF 3D printed PEEK and the influence of printed surface structures on cell adhesion, metabolic activity and proliferation have not been reported [6,21,22]. The novelty of this study is not for PEEK material, rather the analyses of FFF 3D printed PEEK surfaces. Therefore, in this study, we fabricated PEEK samples using FFF and modified the surfaces by polishing and grit-blasting to evaluate the influence of surface topography and surface roughness on cell adhesion, metabolic activity, and proliferation.

## 2. Materials and Methods

### 2.1. Sample Preparation

PEEK disk samples (*n* = 200, *Φ* = 14 × 2 mm^3^) were fabricated with a FFF 3D printer (Apium P220, Apium Additive Technologies GmbH, Karlsruhe, Germany) using the company’s proprietary adaptive heat-controlling software (Apium Additive Technologies GmbH, Karlsruhe, Germany). Table 1 lists the printing parameters used in this study. The filament used was a medical-grade PEEK 3D filament (Evonik VESTAKEEP® i4 G resin, Evonik Industries AG, Essen, Germany; Table 2). Prior to printing, a special fixative spray (DimaFix, DIMA 3D, Valladolid, Spain) was applied to the print bed to achieve adequate adhesion of the disks. The molten thermoplastic PEEK was extruded through the extruder’s nozzle and deposited in a layer-by-layer manner. Once a single layer was deposited, the build platform of the printer moved down according to the layer thickness chosen for the fabrication and deposited the next layer. Because the material was melted when it was extruded, it bonded to the layers deposited beneath. As each layer deposited, cooled and hardened, the final object was fabricated.

### 2.2. Sample Surface Modification

Three kinds of PEEK sample surfaces were evaluated: untreated (FFF 3D printed samples without surface modifications, *n* = 40), polished (*n* = 40) and grit-blasted disks (*n* = 120; *n* = 40 for each subgroup, grit-blasted with 50, 120, and 250 µm alumina (Al_2_O_3_) particles, respectively). Specimens in the polished and grit-blasted groups were polished with a series of SiC abrasive papers with an increasing grit number (Buehler, Lake, Bluff, IL, USA: 1200, 2500 and 4000 grit). After polishing, the samples in the grit-blasted group were modified by blasting with different Al_2_O_3_ grain-size particles (Cobra, Renfert, Hilzingen, Germany) in order to achieve a relatively rougher surface compared with the polished samples. We used a grit-blasting machine (P-G 400, Harnisch+Rieth, Winterbach, Germany) operated at a distance of 50 mm under 0.1 MPa pressure for 15 s. After surface modifications, all of the specimens were ultrasonically cleaned (Sonorex super RK102H, Bandelin, Germany) with deionized (DI) water and 70% ethanol (15 min for each) and dried with nitrogen for 20 s. Finally, the disks were sterilized via autoclave sterilization at 134 °C for 5 min (WESA, Brussels, Belgium).

### 2.3. Surface Morphology, Roughness, and Hydrophilicity

The surface morphologies and microstructures of the different groups of FFF 3D printed PEEK samples were characterized using scanning electron microscopy (SEM; LEO 1430, Zeiss, Oberkochen, Germany) at a magnification of 200×, 2000× and 10,000×. Prior to SEM examination, the sample surfaces were sputtered with Au–Pd coating 20 nm thick (SCD 050, Baltec, Lübeck, Germany).

The sample surface roughness and topography of the different groups of disks were investigated using profilometry (Perthometer Concept S6P, Mahr, Göttingen, Germany; *n* = 6 per group). For the surface topography investigation, 121 profiles perpendicular to the macroscopic printing lines over an area of 3 mm × 3 mm were measured on each sample surface. Two-dimensional (2D) roughness parameters were calculated using analytical software (MountainsMap Universal 7.3, Digital Surf, Besancon, France) with a high-pass robust Gaussian filter (600 µm, ISO 16610-71) in order to separate roughness from waviness [26]. Height parameters, average roughness (Ra), and root-mean-square average roughness (Rq) were analyzed to evaluate and compare surface roughness.

Surface wettability was quantified by measuring the static water contact angle using a drop shape analyzer by setting 2 µL sessile drops of ultrapure water on each sample surface (DSA 10-MK 2, Kruess, Hamburg, Germany; *n* = 6 per group). The equilibrated contact angle of the air-water-substrate interface was analyzed after 20 s of wetting using DSA calculation software (version 1.90.0.11, Kruess, Hamburg, Germany).

### 2.4. Osteoblast Responses to Sample Surfaces

#### 2.4.1. Cell Culture

A human osteosarcoma cell line (SAOS-2 osteoblast, DSMZ GmbH, Braunschweig, Germany) originating from human osteogenic sarcoma was used for this experiment. The cells were cultured in flasks (CellBind T-75, Corning, Tewksbury, MA, USA) in McCoy’s 5A medium (Sigma-Aldrich, Steinheim, Germany) supplemented with 15% fetal bovine serum (FBS, Life Technologies Co., Carlsbad, CA, USA), 1% L-glutamine (GlutaMAX, Life Technologies Co.) and 1% penicillin and streptomycin (15140-122, Life Technologies Co.) in a cell incubator at a culturing temperature of 37 °C in an atmosphere of 5% CO_2_ and 95% humidity. The culture medium was renewed twice a week. The adherent cells on the flask bottoms were separated by 1.5 mL trypsin- EDTA (0.05% trypsin/0.02% EDTA, Life Technologies Co.) for 5 min at 37 °C in an incubator. 

#### 2.4.2. Initial Cell Adhesion

After being cleaned and sterilized, the FFF 3D printed PEEK disks were placed and fixed in the middle of a 24-well culture plate (Cluster, Corning, Tewksbury, MA, USA) with sterile wax. SAOS-2 osteoblasts at a density of 1.5 × 10^5^ cells/cm^2^ were cultured onto the samples (*n* = 8 per group: 4 samples and 4 background controls per group). After 4 h of cultivation, each sample was rinsed with 500 µL Hank’s Salt Solution (HBSS, Biochrom AG, Berlin, Germany) to remove loosely attached cells. The adhering cells were later on fixed with 500 µL 3% paraformaldehyde (MERCK, Haar, Germany) in Dulbecco’s Phosphate-Buffered Saline (DPBS, without calcium and magnesium, Gibco, Paisley, UK) for 15 min at room temperature. After fixation, the paraformaldehyde solution was removed and discarded. Each sample was stained with 500 µL crystal violet dye (Sigma-Aldrich) for 15 min at room temperature. After staining, the samples were rinsed with 500 µL of distilled water five times and transferred to new plates. The samples’ surfaces were photo-documented using a microscope (M400, Wild Heerbrugg, Gais, Switzerland) equipped with a digital camera (EOS 500D, Canon, Tokyo, Japan). Subsequently, the cell-staining dye was solubilized with 500 µL pure methanol (MERCK, Haar, Germany) for 15 min and the absorbance of adhered cells on the sample surfaces was measured at 550 nm using an ELISA photometer (Tecan F50, Tecan Austria, Groedig, Austria). The mean optical density (OD) values of the background control group were subtracted from the corresponding groups. The initial cell adhesion test was performed two times in independent experiments.

#### 2.4.3. Cell Proliferation Assay

Four samples from each group (untreated, polished, 50, 120, and 250 µm Al_2_O_3_ grit-blasted) were fixed in the middle of the 24-well culture plate bottom with sterile wax. SAOS-2 osteoblasts were seeded at a density of 3 × 10^4^ cells/cm^2^ in 1.2 mL modified McCoy’s 5A medium. After being incubated for 1, 3, and 5 days, 600 µL of the medium was removed from each well and 60 µL cell counting kit-8 assay (CCK-8) labeling reagent (Dojindo Molecular Technologies, Inc., Rockville, MD, USA) was added to determine the mitochondrial dehydrogenase activity as a measure for proliferation of the SAOS-2 cells. After incubation for an additional 3 h, the OD value was measured spectrophotometrically using an ELISA reader (Tecan F50, Tecan Austria) at 492 nm (reference wavelength: 620 nm). The culture medium was renewed at the end of each measurement. The test was performed four times in independent experiments.

#### 2.4.4. Cell Density after Prolonged Cultivation

At the end of the last proliferation measurement (i.e., after 5 days), the cell density on the experimental sample surfaces in the proliferation assay was measured. Staining, photo-documentation, and quantification were similar to the procedure described above (Section 2.4.2). The test was performed four times in independent experiments.

### 2.5. Statistical Analysis

SPSS Version 25 (SPSS INC, Chicago, IL, USA) was used to analyze the data. The data distribution and homogeneity of variances were analyzed using the Shapiro–Wilk and Levene tests. One-way analysis of variance (ANOVA) was used to compare the differences among different groups (untreated, polished, 50, 120, and 250 µm Al_2_O_3_ grit-blasted), followed by Tukey post-hoc test (α = 0.05). Unless otherwise indicated, the results are presented as means ± standard deviations and *p*-value < 0.05 are regarded as being statistically significant.

## 3. Results

### 3.1. Surface Characterization

Figure 1 shows the SEM images of the FFF 3D printed PEEK sample surfaces in different groups. Layered patterns on the printed surface comprising of distinct peaks and valleys were detected on the untreated PEEK samples due to the FFF fabrication process (Figure 1a–c). After polishing, the specific printing lines disappeared completely, leaving a homogenous, smoother surface (Figure 1d–f). The grit-blasted surfaces did not exhibit the layered pattern either. Compared with the polished surfaces, the grit-blasted surfaces exhibited a microroughened surface topography with a homogeneous distribution and coverage of protuberances and cavities (Figure 1g–o). Moreover, the samples blasted with the 250 µm grit exhibited an increased level of microroughness on their surface with larger protuberances and cavities compared with the 50 µm and 120 µm Al_2_O_3_-blasted groups, respectively.

### 3.2. Surface Roughness

Figure 2 shows the reconstructed 3D surface topographies and the quantitative roughness results of different groups. FFF 3D printing structures were observed on the untreated PEEK surfaces with Ra and Rq values of 22.28 ± 15.26 µm and 26.75 ± 17.17 µm, respectively. Compared with the untreated group, the polished and grit-blasted samples exhibited significantly smoother and more homogenous surfaces (*p* < 0.05) with lower Ra (polished: 0.17 ± 0.08 µm; 50 µm grit-blasted: 0.28 ± 0.13 µm; 120 µm grit-blasted: 0.43 ± 0.15 µm; 250 µm grit-blasted: 0.52 ± 0.38 µm) and Rq (polished: 0.30 ± 0.15 µm; 50 µm grit-blasted: 0.49 ± 0.25 µm; 120 µm grit-blasted: 0.76 ± 0.23 µm; 250 µm grit-blasted: 0.88 ± 0.56 µm) values.

### 3.3. Surface Hydrophilicity

Surface hydrophilicity was determined by applying 2 µL of ultrapure water to the samples’ surfaces. The results are shown in Figure 3. All of the groups exhibited poor wetting, irrespective of the surface treatments. The contact angles on the different sample surfaces were similar and did not differ statistically (*p* > 0.05, untreated samples: 84.6 ± 9.6°, polished samples: 86.5 ± 4.4°, 50 µm grit-blasted samples: 88.7 ± 3.0°, 120 µm grit-blasted samples: 88.0 ± 2.2°, 250 µm grit-blasted samples: 87.1 ± 3.5°). Moreover, the standard deviation associated with the untreated group were higher than those of the polished and grit-blasted groups.

### 3.4. Initial Cell Adhesion

After seeding the osteoblasts for 4 h, the initial cell adhesion was analyzed by staining with crystal violet. Representative examples of the surface coverage of osteoblasts on exemplary samples and quantitative OD values referring to untreated PEEK are shown in Figure 4. The surface coverages by SAOS-2 osteoblast cells after 4 h in different groups were quite similar across the samples; the exception was the 250 µm grit-blasted group. Surfaces in this group exhibited slightly higher osteoblastic surface coverage than the other groups. Our quantitative results confirmed this finding: the samples grit-blasted with 250 µm Al_2_O_3_ particles exhibited a higher relative OD value (137 ± 45%) compared with the other groups (untreated: 100 ± 10%, polished: 101 ± 14%, 50 µm grit-blasted: 107 ± 13% and 120 µm grit-blasted: 118 ± 21%). The differences between the 250 µm grit-blasted group and the untreated, polished, and 50 µm grit-blasted groups were statistically significant (*p* < 0.05). In addition, the standard deviation of initial cell adhesion for the 250 µm grit-blasted group was also higher than that of the other groups.

### 3.5. Cell Proliferation

To determine the effect of FFF printing structures and surface roughness on the growth of osteoblasts, we used the CCK-8 assay to examine cell metabolic activity as an indirect measurement of cell proliferation (Figure 5). Metabolic activity increased on all of the sample surfaces over the course of 5 days, indicating a continuous proliferation in all the groups. Moreover, there was a significant increase in the CCK-8 reduction activity of the cells on the untreated PEEK sample surfaces (*p* < 0.05) compared with the polished and grit-blasted surfaces. After day 1, a slightly higher OD value (0.69 ± 0.07) indicated faster proliferation of osteoblasts cultivated on untreated surfaces compared with osteoblasts cultivated on the polished (OD value: 0.50 ± 0.05) and grit-blasted surfaces (OD values: 50 µm group: 0.59 ± 0.12; 120 µm group: 0.48 ± 0.10; 250 µm group: 0.59 ± 0.14). As the culturing time increased, the differences became more pronounced. The cells on the untreated surface exhibited higher growth rates after 3 and 5 days (≈2–3 fold) compared with the cells on the polished and grit-blasted surfaces.

### 3.6. Cell Density after Prolonged Cultivation

The cell density on the FFF 3D printed PEEK sample surfaces was determined 5 d after seeding by staining the cell layer with crystal violet. Following photo documentation, the osteoblast density was quantified by eluting the crystal violet and then photometrically quantifying the eluted stain. Figure 6 shows representative examples of cell density after prolonged cultivation of osteoblasts on exemplary samples and the mean cell density on the treated surfaces in comparison to the untreated group (% of the untreated surface). Untreated surfaces of the FFF 3D printed PEEK samples exhibited a significantly enhanced cell density after 5 days compared with the polished and grit-blasted groups. Furthermore, slightly enhanced osteoblastic density was observed on the polished sample surfaces compared with the grit-blasted samples. After prolonged cultivation, quantitative OD values confirmed the finding in microscopic images that the untreated PEEK surfaces had a significantly higher density of attached cells compared with the other groups (*p* < 0.05). The percentage of cells on grit-blasted PEEK was lower than that on the polished surfaces (*p* < 0.05). However, the standard deviation associated with the polished group was higher than that of the grit-blasted groups.

## 4. Discussion

FFF 3D printed PEEK is becoming popular for clinical cranio-maxillofacial and orthopedic applications due to its inherent advantages and the ability to produce complex geometries [6]. Although several publications have focused on the mechanical properties, crystallinity and biocompatibility properties of FFF 3D printed PEEK, studies related to its biological activity—including cell adhesion, cell metabolic activity, and cell proliferation—are still lacking [6,10,22]. Therefore, this investigation aimed to study whether different modifications of the surface structure of PEEK can promote osteoblast adhesion and subsequent proliferation, resulting in faster colonization on the implant surface. Cellular behavior of SAOS-2 osteoblasts was studied on PEEK surfaces of various roughness and compared with the original printed surface. This study highlights four critical issues in cell-material interactions: cell adhesion, metabolic activity, proliferation, and attachment after prolonged incubation periods.

In this study, FFF 3D printed PEEK samples underwent different surface modifications, including polishing and grit-blasting. Grit-blasting is a common surface processing method used in dentistry to roughen sample surfaces [27,28]. However, the untreated group in this study had an anisotropic surface structure with unique printed line and valley structures formed by the FFF manufacturing process. This situation resulted in higher roughness values than those of the polished and grit-blasted groups (*p* < 0.05). This structure is due to the working principles of the FFF technology [27]. The surface topography observed on the samples’ surfaces arose from the manufacturing principles of FFF technology, which involves extruding and bonding of the molten PEEK filament [2]. The extruding PEEK thermoplastic generates a 2D layer with a specific layer thickness, and the successive 2D layers fuse together to build up the final 3D object. As a result of this process, some unfilled areas persist between each 2D layer, which results in the original waveforms on the printed surface [27,29]. In this study, the roughness values (Ra and Rq) of the grit-blasted groups were numerically higher than those of the polished group. However, the roughness values of the grit-blasted groups were not statistically different between polished and grit-blasted groups. The Ra values for both the polished and grit-blasted groups were between 0.17 and 0.52 μm, lower than the results from other studies as well as the commercially available Ti implants [16,30,31,32]. The explanation for the low roughness values between the polished and grit-blasted groups is the grit-blasting parameters selected for this study. Because the PEEK samples were FFF printed rather than traditionally manufactured (i.e., via injection molding), the interior of the object retained some unfilled area based on the infill parameter chosen [6]. Using a higher pressure, a closer distance or a longer duration of the grit-blasting process may have resulted in the exfoliation of the upper layers from the sample surfaces (Figure 7) [10]. Therefore, based on the selected grit-blasting parameters for this study, the grit-blasted PEEK indicated only slightly higher roughened surfaces compared with the polished group. In further studies, other surface treatment methods, e.g., plasma and UV treatment, might be explored to modify the FFF 3D printed PEEK surface [17]. 

Like surface topographical analysis, wettability measurements are also critical for implants because surface wettability influences the adhesion of adsorbed plasma proteins and macromolecules for cell adhesion, proliferation, and differentiation [33,34]. In this study, PEEK samples in all the groups (untreated, polished, and grit-blasted) were characterized by low wettability (Figure 3); there were no statistically significant differences in contact angle measurements among the groups. This result illustrated that the topographical modification methods used in this study did not change the surface wettability significantly. According to Elawadly et al., the water contact angle of PEEK material decreases with an increase of surface roughness within a certain limit, and this relation does not persist beyond a certain point [35]. Furthermore, the wettability decreases when the Ra values are either below 1.0 µm or above 1.7 µm. The water droplets are not able to spread easily on either mostly smooth or highly roughened surfaces. On smoother surfaces, peaks and valleys are not sufficiently pronounced for droplets to spread [35]. On highly roughened surfaces, high peaks and deep valleys act as “geometrical barriers” that prevent droplets from spreading [36]. Deep valleys, on the other hand, provide opportunities for the creation of large-volume air gaps between the sample’s surface and the liquid droplet, thereby increasing the water contact angle [30]. This situation might explain why in this study the contact angle values among the different groups showed a comparable low wettability. To prevent exfoliation of the upper layers of the FFF 3D printed PEEK samples, low-intensity grit-blasting parameters were adopted in this study. As a result, the Ra values of the polished and grit-blasted groups were between 0.17 and 0.52 μm, which is not ideal for surface hydrophilicity [35]. The smooth surface of the polished and grit-blasted groups did not possess enough peaks and valleys for the droplets to spread. The untreated group surfaces, on the other hand, were highly rough and therefore exhibited huge peaks and valleys that formed “geometrical barriers” and prevented the droplets from spreading.

The initial cell adhesion of osteoblasts influences cellular functions and eventually bone integration; cell proliferation is correlated with new bone formation [32]. Better cell adhesion and proliferation yield a larger amount of bone tissue around implants, and stronger bone-implant bonding can also be expected in vivo [25]. In this study, the cell adhesion of SAOS-2 osteoblasts on sample surfaces was analyzed by staining with crystal violet dye after 4 h, which was deemed sufficient to reflect the cell adhesion during the early “decisive period” [16]. The results revealed that the number of adhered cells on each sample surface was similar, except for the 250 μm grit-blasted group (Figure 4). The observed low wettability might be an explanation for the similar initial cell adhesion results from the different groups. As noted above, it has been recognized that the surface wettability of biomaterials is important for their biofunction [16]. Surface wettability might affect a biological system in four major ways: (a) adhesion of proteins and macromolecules onto the surface; (b) cell interactions of hard and soft tissues with preconditioned surfaces; (c) bacterial adhesion and biofilm formation; and (d) in vivo rate of osseointegration [32]. Successful bone-implant osseointegration requires surface colonization of osteoblast precursor cells, followed by proliferation and differentiation, extracellular matrix synthesis and tissue reconstruction. Hydrophilicity influences the initial processes of cell adhesion and spreading, which also has an impact on the subsequent steps in the healing stages, ultimately leading to long-term osseointegration [32]. Huang et al. demonstrated the connection between hydrophilicity and the adhesion of different cells [37]. These authors showed that cell adhesion and spreading were almost completely inhibited on hydrophobic surfaces; enhanced cell adhesion was observed in hydrophilic regions. Ranella et al. analyzed roughness and wettability on cell adhesion and found that a pronounced change in wettability could switch a surface’s behavior from cell-phobic to cell-philic [38]. The wettability results of this study were strikingly similar among the different groups, which might lead to approximate cell adhesion and spreading. Compared with surface roughness, the influence of wettability was more obvious on the initial cell adhesion.

Surface roughness and morphology play an important role in the bioactivities of biomaterials. However, the relationship between surface morphology and bioactivity behavior remains unclear [25]. In this study, we investigated cell metabolic activity on the 1st, 3rd, and 5th days after incubation to explore cell proliferation responses on different PEEK sample surfaces. As shown in Figure 5a,b, the osteoblasts on the untreated sample surfaces exhibited slightly higher metabolic activity than the osteoblasts in the other groups after 1 day, indicating an enhanced proliferation on untreated surfaces. After 3 days, the difference between the untreated group and other groups was more pronounced. The cell metabolic activity of the untreated group was twice than that of the polished group and three times than that of the grit-blasted groups after 5 days. The results indicated that after one day SAOS-2 osteoblasts grew rapidly on the highly roughened surfaces (untreated group); their proliferation rate was much higher than that of the polished and grit-blasted groups. The reasons why FFF 3D printed highly roughened surfaces support the metabolic activity and proliferation of osteoblasts are not entirely clear. The most likely explanation for the increased cell bioactivity on untreated surfaces may be an increase in the available surface area. The availability of a larger surface area usually results in higher cell interactions. [39]. After seeding, the initial cell adhesion between different groups was similar. With prolonged culturing time, however, the highly roughened surfaces of the untreated PEEK with an enlarged surface area resulted in a boosted cell proliferation rate. Large surface areas allow for better cell attachment, anchorage, growth, migration, and proliferation [39]. In this study, the Ra value of the untreated group was about 22.28 µm, which means that the dimension of the peaks and the valleys on this surface is similar to the cell size. A high roughness value suggests a larger surface area for the untreated surface, which might have provided better cell proliferation. In addition, cells can slide into the valleys on the sample surfaces and attach, resulting in a dense cell layer from the beginning of the experiment. This situation might result in an accelerated conditioning of the microenvironment around these cell accumulations, which in turn might have stimulated cell proliferation. The faster proliferation on the untreated sample surfaces may be caused by a combination of enhanced conditioning of the surrounding by the cells in the grooves due to a tighter connection (quasi an auto-stimulation) in the beginning, along with a faster proliferation of this cell type on smooth surfaces. As shown in Figure 8, cell accumulation in the surface grooves resulting from the FFF manufacturing process was observed [10]. 

As a final test, performed after 5 days of incubation, the cell density on the sample surfaces was analyzed with crystal violet staining. This test, which reveals cell size and number, was selected because it provides a qualitative optical evaluation of the attachment of osteoblasts, as well as quantitative assessment by photometric measurement of stained cells. After 5 days, an entire cell-to-cell and cell-to-extracellular-matrix interconnecting network on the sample surfaces was built (Figure 6). The untreated PEEK surfaces had significantly higher osteoblastic cell density compared with the surfaces of the polished and grit-blasted groups. This finding is consistent with the tests of metabolic activity measurements. Additionally, the surfaces in the polished group exhibited slightly higher modified OD values compared with surfaces in the grit-blasted groups. One explanation for this finding is that, for the grit-blasted samples, there might have been some residual alumina left on the sample surfaces despite the samples being cleaned ultrasonically with DI water and ethanol after grit-blasting. At later stages of culturing, the presence of alumina particles and the coarsely blasted morphology might have impeded cell spreading, which impaired the cell proliferation [40]. 

## 5. Conclusions

We systematically analyzed the bioactivity of FFF 3D printed PEEK by investigating surface roughness and wettability, cell adhesion, metabolic activity, and proliferation. The results revealed that the FFF manufacturing technique could yield highly roughened surfaces and special printing structures not achieved with traditional grit-blasting methods. Wettability plays an important role in initial cell adhesion; as the culture time lengthened, the influence of the sample surface morphology and roughness became increasingly apparent. Compared with the polished and grit-blasted PEEK samples, the anisotropic texture of the surfaces printed by FFF had a stimulating effect on the bioactivity of PEEK samples, especially in terms of cell metabolic activity and proliferation. 

Our in vitro tests revealed that FFF 3D printed PEEK with its anisotropic printing structures and surface roughness is a promising material for improving cell adhesion, metabolic activity, and proliferation. Therefore, FFF 3D printed PEEK could be a potential candidate biomaterial for dental and cranio-maxillofacial implants. 

## Figures and Tables

**Figure 1 jcm-08-00771-f001:**
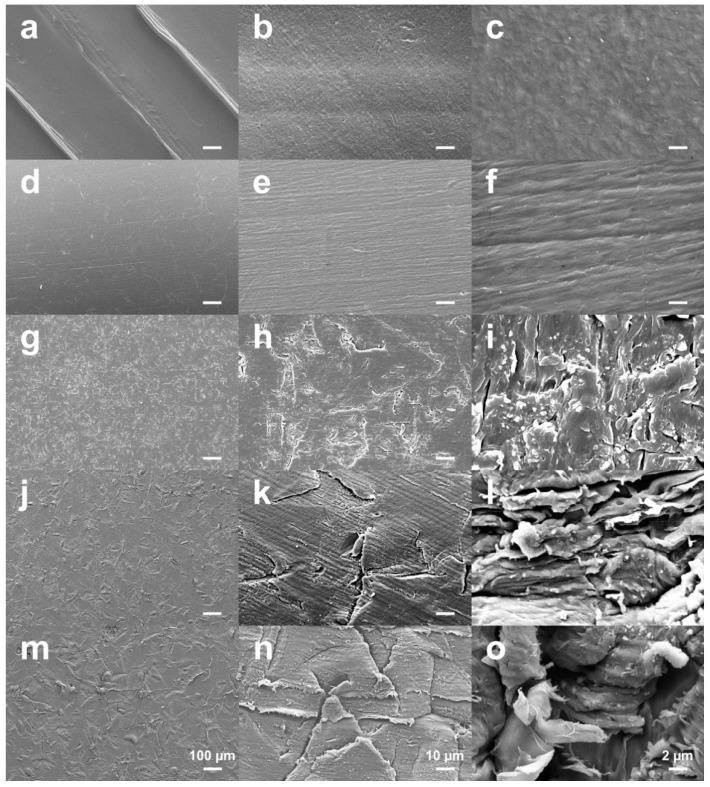
SEM images of the FFF 3D printed PEEK sample surfaces in different groups: (**a**–**c**) untreated; (**d**–**f**) polished; (**g**–**i**) grit-blasted (50 µm); (**j**–**l**) grit-blasted (120 µm); (**m**–**o**) grit-blasted (250 µm). Bars represent 100, 10, and 2 µm from left to right, respectively.

**Figure 2 jcm-08-00771-f002:**
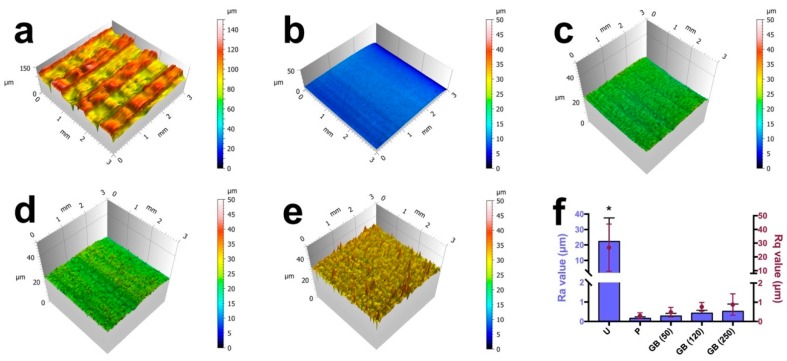
Reconstructed 3D surface roughness topographies of the FFF 3D printed PEEK samples: (**a**) untreated; (**b**) polished; (**c**) grit-blasted (50 µm); (**d**) grit-blasted (120 µm); (**e**) grit-blasted (250 µm). (**f**) Ra and Rq values of the different groups. The data are presented as means ± standard deviations, * *p* < 0.05. U: untreated group; P: polished group; GB (50): 50 µm Al_2_O_3_ grit-blasted group; GB (120): 120 µm Al_2_O_3_ grit-blasted group; GB (250): 250 µm Al_2_O_3_ grit-blasted group.

**Figure 3 jcm-08-00771-f003:**
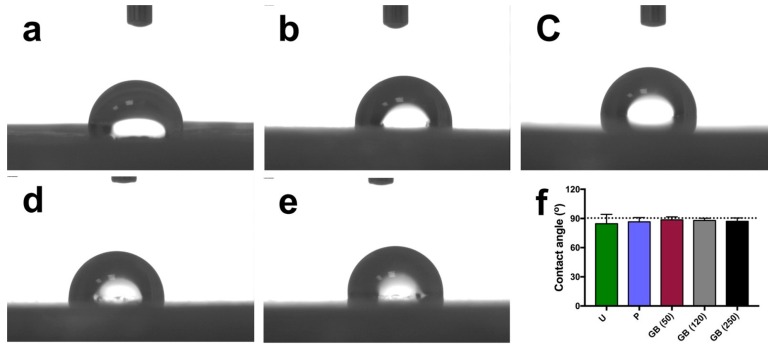
Contact angle measurements of different FFF 3D printed PEEK samples. (**a**) untreated; (**b**) polished; (**c**) grit-blasted (50 µm); (**d**) grit-blasted (120 µm); (**e**) grit-blasted (250 µm); (**f**) quantitative contact angle values (means ± standard deviations). The dotted line indicates a contact angle of 90°, the borderline between hydrophilicity and hydrophobicity. U: untreated group; P: polished group; GB (50): grit-blasted group with 50 µm Al_2_O_3_ particles; GB (120): grit-blasted group with 120 µm Al_2_O_3_ particles; GB (250): grit-blasted group with 250 µm Al_2_O_3_ particles.

**Figure 4 jcm-08-00771-f004:**
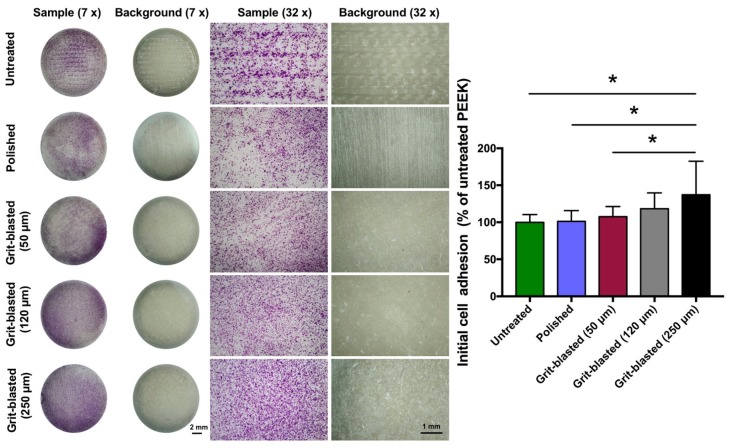
Initial cell adhesion test of the FFF 3D printed PEEK samples in different groups. Representative images of crystal violet-stained osteoblasts on the sample surfaces after 4 h in culture. Left: overview, mag. 7×; right: detail, mag. 32×. The bar chart shows the relative absorbance of the different groups (normalized by the untreated FFF 3D printed PEEK). The data are presented as means ± standard deviations, * *p* < 0.05.

**Figure 5 jcm-08-00771-f005:**
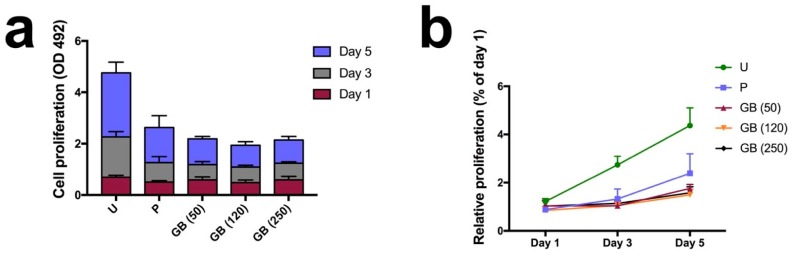
Cell proliferation on the sample surfaces at different time points assessed via CCK-8 assay. (**a**) cell proliferation (OD value at 492 nm wavelength); (**b**) relative proliferation on the 3rd and 5th day, normalized by the values on the 1st day. U: untreated group; P: polished group; GB (50): 50 µm grit-blasted group; GB (120): 120 µm grit-blasted group; GB (250): 250 µm grit-blasted group.

**Figure 6 jcm-08-00771-f006:**
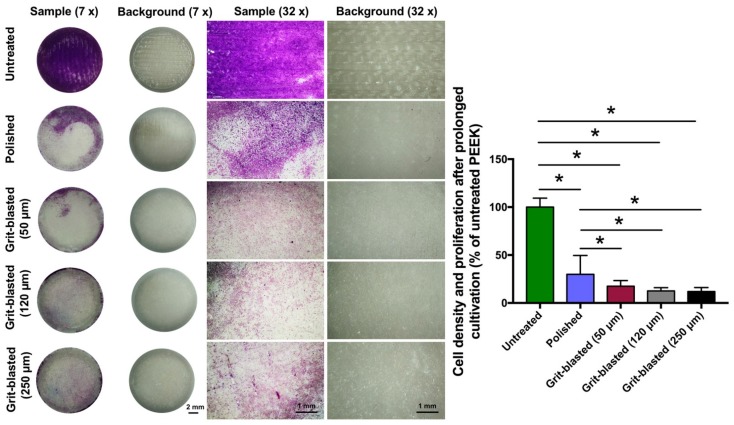
Cell density of SAOS-2 for the different groups after 5 days of cultivation. Left: overview, mag. 7×; right: detail, mag. 32×. The bar chart shows the relative absorbance of different groups (normalized by untreated PEEK). The data are presented as means ± standard deviations, * *p* < 0.05.

**Figure 7 jcm-08-00771-f007:**
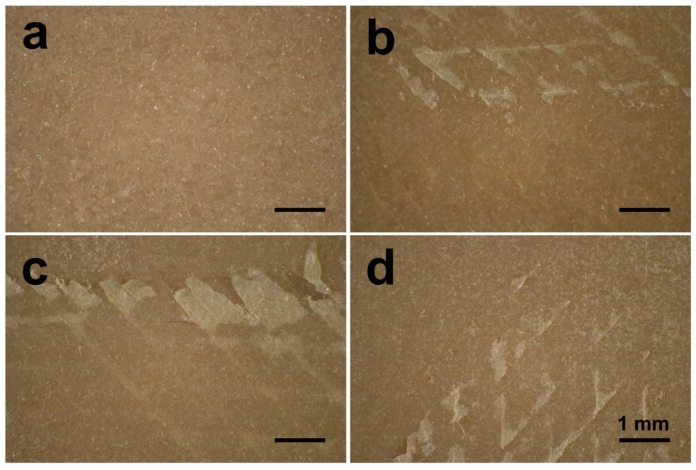
Optical micrographs of the grit-blasted PEEK samples after different grit-blasting procedures: (**a**) distance: 50 mm, pressure: 0.1 MPa, time: 15 s; (**b**) distance: 10 mm, pressure: 0.1 MPa, time: 15 s; (**c**) distance: 50 mm, pressure: 0.5 MPa, time: 15 s; (**d**) distance: 50 mm, pressure: 0.1 MPa, time: 60 s [10].

**Figure 8 jcm-08-00771-f008:**
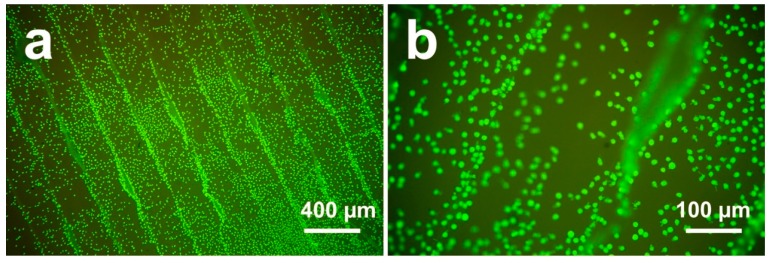
LIVE/DEAD staining of cells attached to FFF 3D printed PEEK sample surfaces after culturing for 24 h: (**a**) 25× magnification; (**b**) 100× magnification [10].

**Table 1 jcm-08-00771-t001:** The fabricating parameters of the FFF 3D printer.

Description	Value
Layer thickness	0.2 mm
Nozzle diameter	0.4 mm
Filament diameter	1.75 mm
Print head	480 °C
Print bed	130 °C

**Table 2 jcm-08-00771-t002:** The material properties of the polyetheretherketone (PEEK) filament.

Description	Value
Glass-transition temperature Tg	143 °C
Melting temperature Tm	343 °C
Density	1.30 g cm^−3^
Tensile strength	96 MPa
Tensile modulus	3500 MPa

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
