# Peer review of "An In Vitro Study of Osteoblast Response on Fused-Filament Fabrication 3D Printed PEEK for Dental and Cranio-Maxillofacial Implants"

_jcm, 2019, doi:10.3390/jcm8060771_

Reviewer 1 Report

In the present study, the biocompatibility of 3D-printed PEEK toward human osteosarcoma cell line Saos-2 is investigated. Authors fabricated PEEK discs using 3D technology, threated these discs with different methods in order to produce surfaces with different characteristics and investigated initial attachment and growth of osteosarcoma cells on these surfaces. The study is well planned and performed, experimental procedures and data are clearly presented in the most cases. However, there are some issues, which need to be revised by Authors.

1.       Authors should clearly define what they measure and which processes underline the measured values. For example, Fig. 5 shows the data of CCK-8 assay. This assay is based on the measurements of cell mitochondrial activity and reflects both cell viability and proliferation. The data cannot be interpreted as “Cell metabolic activity and proliferation”, because these two factors cannot be separated and using of this term might mislead the readers.

2.       Measurements of cell coverage (Fig. 6). The use of this term is also controversial. According to the applied procedure, cells were fixed and stained after some period of culture, and the amount of stained color was quantified. Thus, the intensity of color should be proportional to the number of the cells and thus reflect cell proliferation.

3.       The differences in Ra parameter between polished and grit-blasted surfaces are rather small. Moreover, Ra values of grit-blasted surfaces are markedly lower compared to those of commercially available Ti implants. This fact should be considered in Discussion.

4.       Ra parameter of untreated PEEK discs was pretty high (22.28 µm). It means that the dimension of the hills and the valleys on this surface is similar to the cell size. This parameter does not really reflect biologically relevant surface microstructure. It is known that cell behavior is influenced by moderately rough surfaces with Ra in the range up to 4 µm. However, surface with so high Ra values will influence cell growth by other mechanisms. Moreover, such high roughness values suggest much larger surface area, and this might account better proliferation observed for this surface.

Author Response

Thank you very much for reviewing our manuscript. Your suggestions are very helpful for us to improve our paper. We have already revised our manuscript according to your comments, especially for some details you pointed out.

Reviewer 2 Report

The work does not have novelty. The group used ready filaments even without modifying PEEK. In addition, the authors only used the polymer without using any filler to impart antibacterial activity or increasing the mechanical stability. at least a filler for increasing the mechanical properties should be added. Only surface finishing (surface treatment) is not enough.  My other suggestions:

Table 1 must be removed as it is not necessary (similar to advertisement).

 The number of analysis is not bad. but it need mechanical tests. They suggested to use instead of Titanium but no mechanical test was shown.

The figures are good.

This ref could be cited in the introduction.

Antimicrobial modified hydroxyapatite composite dental bite by stereolithography. Polymers for Advanced Technologies 2017.

The number of references is high. around 40 would be enough.

Author Response

Thank you very much for reviewing our manuscript. Your comments are helpful for us to improve our paper. We have already revised our manuscript according to your suggestions.

Round  2

Reviewer 2 Report

The authors modified the manuscript. Now, it seems it is better than before and may be acceptable for publications.